# Catenin Alpha 2 May Be a Biomarker or Potential Drug Target in Psychiatric Disorders with Perseverative Negative Thinking

**DOI:** 10.3390/ph14090850

**Published:** 2021-08-26

**Authors:** Nora Eszlari, Zsolt Bagyura, Andras Millinghoffer, Tamas Nagy, Gabriella Juhasz, Peter Antal, Bela Merkely, Gyorgy Bagdy

**Affiliations:** 1Department of Pharmacodynamics, Faculty of Pharmacy, Semmelweis University, Nagyvárad tér 4, H-1089 Budapest, Hungary; juhasz.gabriella@pharma.semmelweis-univ.hu (G.J.); bagdy.gyorgy@pharma.semmelweis-univ.hu (G.B.); 2NAP-2-SE New Antidepressant Target Research Group, Hungarian Brain Research Program, Semmelweis University, Nagyvárad tér 4, H-1089 Budapest, Hungary; milli@mit.bme.hu; 3Heart and Vascular Center, Semmelweis University, Városmajor utca 68, H-1122 Budapest, Hungary; bagyura@zortal.hu (Z.B.); merkely.bela@med.semmelweis-univ.hu (B.M.); 4Department of Measurement and Information Systems, Budapest University of Technology and Economics, Magyar Tudósok krt. 2, H-1521 Budapest, Hungary; tnagy@mit.bme.hu (T.N.); antal@mit.bme.hu (P.A.); 5Abiomics Europe Ltd., Zólyomi út 23, H-1118 Budapest, Hungary; 6MTA-SE Neuropsychopharmacology and Neurochemistry Research Group, Hungarian Academy of Sciences, Semmelweis University, Nagyvárad tér 4, H-1089 Budapest, Hungary; 7SE-NAP 2 Genetic Brain Imaging Migraine Research Group, Hungarian Brain Research Program, Semmelweis University, Nagyvárad tér 4, H-1089 Budapest, Hungary

**Keywords:** *CTNNA2*, catenin alpha 2, rumination, perseverative cognition, Framingham Risk Score, body mass index, Brief Symptom Inventory, psychiatric symptoms

## Abstract

AlphaN-catenin gene *CTNNA2* has been implicated in intrauterine brain development, as well as in several psychiatric disorders and cardiovascular diseases. Our present aim was to investigate *CTNNA2* gene-wide associations of single-nucleotide polymorphisms (SNPs) with psychiatric and cardiovascular risk factors to test the potential mediating role of rumination, a perseverative negative thinking phenotype in these associations. Linear mixed regression models were run by FaST-LMM within a sample of 795 individuals from the Budakalasz Health Examination Survey. The psychiatric outcome variables were rumination and its subtypes, and ten Brief Symptom Inventory (BSI) scores including, e.g., obsessive-compulsive, depression, anxiety, hostility, phobic anxiety, and paranoid ideation. Cardiovascular outcome variables were BMI and the Framingham risk scores for cardiovascular disease, coronary heart disease, myocardial infarction, and stroke. We found nominally significant *CTNNA2* associations for every phenotype. Rumination totally mediated the associations of *CTNNA2* rs17019243 with eight out of ten BSI scores, but none with Framingham scores or BMI. Our results suggest that *CTNNA2* genetics may serve as biomarkers, and increasing the expression or function of CTNNA2 protein may be a potential new therapeutic approach in psychiatric disorders with perseverative negative thinking including, e.g., depression. Generally, an antiruminative agent could be a transdiagnostic and preventive psychopharmacon.

## 1. Introduction

The *CTNNA2* gene and the encoded catenin (cadherin-associated protein) alpha 2 (αN-catenin) protein have a crucial role in brain development. It has been suggested to regulate the branching of actin. Actin, in turn, guides the cytoskeleton’s microtubules within growing neurites of developing and migrating neurons of the fetal cortex [1].

This risk gene acting at an early, intrauterine stage of brain development, has been associated with numerous psychiatric and psychological phenotypes in humans. It has been implicated in bipolar disorder [2], adult attention-deficit/hyperactivity disorder (ADHD) [3], alcoholism [4], schizophrenia, and general cognitive function [5], as well as in success in smoking cessation [6]. Moreover, it has shown associations with personality traits and endophenotypes that lie on possible pathways between genes and disorders, namely: impulsivity, which covers both novelty seeking and lack of planning [7], as well as a common visual endophenotype of schizophrenia and autism spectrum disorder, namely deficits in sensitivity to visual stimuli of a low spatial and high temporal frequency [8].

Although it has not survived multiple testing correction at a genome-wide level, it has shown a very significant association with Framingham Risk Score, an indicator of cardiovascular disease risk [9], and similarly with orthostatic hypotension [10].

It is a very challenging question whether or not the same mechanism of *CTNNA2* acts in distinct pathways of psychiatric and cardiovascular risk. Regarding the potential psychological endophenotypes that mediate between genes and these distinct types of disorders, rumination would be a good candidate to investigate.

Rumination, or depressive rumination, is a stable response style to the person’s own distress and depressed mood [11,12]. This trait involves a repetitive and passive focus, a fixation on personal problems and feelings, as well as on their causes and consequences, without taking action [11]. Therefore, this perseverative cognition prolongs mental representations, e.g., affective and physiological activation patterns of a stressor, being a moderator between stress and its health consequences [13].

Rumination has indeed been suggested as a transdiagnostic risk factor [14]. It predicts the future onset of major depression, even over many years [15]. It has been found to be present in all phases of bipolar disorder, associated with symptoms of depression, hypomania, and anxiety [16], and has been shown to have a higher level in social anxiety than in non-anxious subjects [17]. Rumination also predicted symptoms of post-traumatic stress disorder (PTSD) one year [18] and even three years [19] after a motor vehicle accident. It also predicted the future onset of substance abuse and binge eating in female adolescents [20], as well as the future level of alcohol use in alcohol abusers [21]. Rumination is also associated with premenstrual disorders [22] and shows its notable maladaptive aspects in schizophrenia as well [23].

Rumination, a form of a maintained “action preparation” stress response because of an affective, attentional, and autonomic inflexibility, has also been suggested to predict somatic complaints one year later [13]. There is also evidence that rumination is associated with a slower recovery of heart rate or blood pressure after stress, and such a slower cardiovascular recovery, in turn, predicts the future development of hypertension, even when controlling for the initial reactivity level [13,24]. A direct relationship between rumination and the future emergence of cardiovascular diseases, as well as the precedence of either cognitive or autonomic inflexibility in these relationships, still has to be elucidated [24].

Our present aim was to test the potential mediating role of rumination and its subtypes in *CTNNA2*-associated psychiatric and cardiovascular risk. For this, first, we tested associations of rumination with psychiatric and cardiovascular phenotypes. After that, we tested the role of gene-wide single-nucleotide polymorphisms (SNPs) of *CTNNA2* in psychiatric symptoms, rumination, and cardiovascular risk factors, including body mass index (BMI) and Framingham scores. Finally, we tested the mediating role of rumination in *CTNNA2*-associated psychiatric and cardiovascular phenotypes.

## 2. Results

### 2.1. Associations between Rumination Score and Disease Risk Phenotypes

The mean, standard error, and standard deviation for each variable, as well as statistical power for the outcome variables and Cronbach’s alpha values for psychometric scales, are presented in Table 1.

While rumination was associated with all ten Brief Symptom Inventory (BSI) scores, no cardiovascular risk factor was associated with it (Table 2). A moderate positive correlation was found in all cases.

#### Associations between Other Ruminative Response Scale (RRS) Scores Brooding and Reflection and Disease Risk Phenotypes

RRS rumination has two subtypes: brooding and reflection [25]. Brooding is a “moody pondering”, passive, and maladaptive comparison of a person’s current situation with some unachieved standard. In contrast, reflection is a more adaptive subtype, which means a neutrally valenced, purposeful turning inward to engage in cognitive problem solving and thus to alleviate the depressed mood.

The RRS score residuals significantly correlated with each other – both brooding and reflection had a positive correlation with rumination, and had a negative correlation with each other. While RRS brooding had a moderate positive association with all ten BSI scores, only BMI among cardiovascular risk factors was associated with brooding—weakly and positively. RRS reflection had weak significant associations, was positive with depressive and obsessive-compulsive symptoms, and was negative with BMI (Table 2).

### 2.2. Associations of CTNNA2 SNPs with Each of the Phenotypes, in FaST-LMM Regression Models

Figure 1 demonstrates that while no SNP within *CTNNA2* or its buffer region extended by ±10 kilobase survived the Bonferroni-corrected significance threshold, many SNPs showed a nominal association with every phenotype. Rs12613937 and rs13030077 were the two most significant SNPs for Framingham-CVD, Framingham-CHD, and Framingham-HCHD.

### 2.3. The Mediating Role of Rumination between CTNNA2 and Psychiatric Symptoms

Mediation testing has two prerequisites. First, the potential mediator has to be significantly associated with the outcome variable (Table 2), and second, the potential mediator and the outcome have to share at least one significant SNP with each other (Appendix A).

As none of the cardiovascular risk scores are correlated with rumination (Table 2), no mediation is possible. Rumination has positive correlations with all ten BSI scores (Table 2) and shares six significant SNPs with at least one of the BSI scores (Appendix A). Consequently, these six SNPs, rumination, and the respective BSI scores meet both the prerequisites of mediation testing. Among the shared SNPs with rumination, associations with eight BSI scores were found with two SNPs: rs12615043 and rs17019243 (Appendix A). In detail, rs12615043 is significantly associated with rumination, as well as with BSI global severity index (GSI), obsessive-compulsive, interpersonal sensitivity, depression, anxiety, phobic anxiety, paranoid ideation, and psychoticism scores. Rs17019243 is significantly associated with rumination, BSI GSI, somatization, obsessive-compulsive, depression, anxiety, hostility, phobic anxiety, and paranoid ideation scores.

To show the potential mediating role of rumination in BSI associations with the six shared SNPs, we ran secondary analyses in FaST-LMM regression models for the ten BSI scores, covarying the rumination score (Appendix A). Most of the SNP-BSI score associations lost their significance if covarying rumination, suggesting a mediating role.

As they had associations with most of the BSI scales, rs12615043 and rs17019243 were included in a Bayesian Multilevel Analysis (BMLA) with all of the 18 investigated phenotypes, plus sex and age, to reveal their relevance regarding RRS and BSI scores within the network of all of the investigated variables. Because of the low sample sizes within the separate genotype groups (rs12615043 GG: 5; GA: 83; AA: 550; rs17019243 TT: 5; TC: 52; CC: 577), a dominant model was used, with binary encoding. The results pointed out that while rs12615043 proved to be irrelevant regarding either the RRS or BSI scores, rs17019243 had high posterior probabilities of a relationship with RRS rumination and brooding, BSI somatization, and the categories of both RRS and BSI (Figure 2 and Appendix A). Supported by the results of this non-frequentist statistical approach, in addition to the results of LMM regression models, rs17019243 was selected for testing mediation via rumination towards BSI scores, where primary analyses showed associations, namely GSI, somatization, obsessive-compulsive, depression, anxiety, hostility, phobic anxiety, and paranoid ideation scores. 

Statistical mediation was investigated by a structural equation model (SEM) analysis including 634 unrelated participants who had non-missing data on rs17019243, sex, age, all three RRS scores, and all ten BSI scores. Rs17019243 was included according to a dominant model—whether or not the participant carried the minor T allele. The following regressions were included to control for the effects of confounding variables. The eight BSI scores and the potential mediator RRS rumination score were regressed on sex and age. Potential genetic stratification of the population also had to be corrected for, so as to avoid false positive genetic associations with the investigated phenotypes. As the first principal component (PC) of the genome showed a correlation with the BSI global severity index and anxiety, and the 10th PC of the genome showed a correlation with BSI obsessive-compulsive and hostility scores—these BSI scores were regressed on the respective PC in the model.

The model showed excellent fit indices (Figure 3). To display the direct effects of rs17019243 for rumination and BSI scores within the model, regression (path) coefficients were standardized by standard deviation of the outcome variable (Figure 3). Similarly, to show the mediating role of rumination within the same model, estimates of the indirect effects were standardized by standard deviations of the outcome variable (Table 3). The results revealed that while rs17019243 had no significant direct effect on any of the psychiatric symptom scores if a mediated effect via rumination was also included in the model, all of these mediated effects were significant (Figure 3 and Table 3). These results are in line with the notion that *CTNNA2* SNP effects on the BSI scores are entirely mediated by rumination.

#### The Mediating Role of Other RRS Scores Brooding and Reflection between *CTNNA2* and Psychiatric Symptoms

The results of the brooding and reflection are shown in the same tables as the rumination (Table 2 and Appendix A).

As none of the Framingham scores were correlated with any of the RRS scores (Table 2), no mediation was possible. Regarding BMI, although it has a weak association with both brooding and reflection scores (Table 2), it shares no significant SNP with any of them (Appendix A), excluding the possibility of mediation. Similarly, in case of RRS reflection, although it was correlated with both BSI obsessive-compulsive and depressive symptoms, it shared no significant SNP with any of them, excluding any mediating role. Regarding associations and positive correlations with BSI scores and the number of shared SNPs, brooding was very similar to rumination (Table 2 and Appendix A). However, among the rs12615043 and rs17019243, brooding had a significant association only with rs12615043 according to LMM primary analyses (Appendix A), which SNP, in turn, did not fulfill the criterion of relevance in BMLA results (Figure 2 and Appendix A), therefore no mediating role of brooding was tested.

## 3. Discussion

Our study was the first to investigate whether SNPs covering the whole *CTNNA2* gene are related to a diverse set of psychiatric and cardiovascular risk scores, as well as whether these associations are mediated by a transdiagnostic repetitive negative thinking phenotype—rumination. Our results demonstrated that while *CTNNA2* SNPs had nominal associations with all psychiatric and cardiovascular risk scores, rumination mediated *CTNNA2*′s role only towards psychiatric, but not cardiovascular risk scores.

### 3.1. CTNNA2 Has Pleiotropic Effects on Cardiovascular Phenotypes and Rumination

*CTNNA2* SNPs showed nominal associations with each of the investigated 18 phenotypes. However, while *CTNNA2* associations with psychiatric risk scores seem to be mediated by rumination, its associations with cardiovascular risk scores seemed independent of rumination.

Although rumination, as an inflexibly maintained stress response and a repetitive thought style, seemed to be a relevant cardiovascular risk [13,24], it was not a significant factor in the *CTNNA2* variant-associated increase of Framingham risk scores.

In fact, *CTNNA2* was found to have associations with BMI and Framingham scores independently of rumination. *CTNNA2* SNPs may act on cardiovascular phenotypes through brain mechanisms other than rumination, as it has been hypothesized in case of *CTNNA2* involvement in orthostatic hypotension [10].

Another possible explanation is that the CTNNA2 protein is present in the heart, and thus has a direct cardiovascular effect. Although, according to the Human Protein Atlas v20.0 [26], *CTNNA2* shows the highest RNA and protein expression levels in brain, male tissues, and granulocytes (https://www.proteinatlas.org/ENSG00000066032-CTNNA2/tissue accessed on: 22 November 2020), it also has some low expression values in different cell types of the heart muscle (https://www.proteinatlas.org/ENSG00000066032-CTNNA2/celltype accessed on 22 November 2020).

Further studies are needed to elucidate the precise nature of this pleiotropy on rumination and risks for coronary heart disease, stroke, cardiovascular diseases generally, or BMI.

### 3.2. CTNNA2 Effects on Divergent Psychiatric Symptoms Are Entirely Mediated by Rumination

Our results revealed that while *CTNNA2* rs17019243 acted on eight BSI scores via RRS rumination, no direct effect could be detected from the SNP towards any of the BSI scores.

Although the genetic background of rumination has been extensively investigated in previous studies [27,28,29], only a few studies have considered the possible endophenotypic nature of rumination, namely its putative causal role on the pathways between specific genes and specific disorders. These few studies, implicating the roles of synaptic plasticity candidate genes *BDNF* [30] and *CREB1* [31], as well as serotonin receptor gene *HTR2A* [28] and folate pathway gene *MTHFD1L* [32], have exclusively focused on depression as a relevant disorder, or on depressive symptoms. No other disorders have been studied as the end of pathways from certain genes through rumination. Nevertheless, a twin study demonstrated a substantial genetic overlap between rumination and depression, though a more complicated picture in the association of rumination and eating pathology, and a firm role of environmental influences instead of genetics in the association of rumination and dependence vulnerability [33]. The BSI questionnaire of our present study did not assess eating pathology or addiction, thus we could not investigate the endophenotypic features of RRS scales between *CTNNA2* and these two kinds of psychopathology. However, our results suggested the endophenotypic relevance of rumination in pathways between *CTNNA2* and several other psychiatric symptom scores, also corroborating former results with depressive symptoms. Furthermore, the transdiagnostic nature of rumination in *CTNNA2*-associated psychiatric disorders cannot be separated or restricted to brooding or reflection subtypes.

### 3.3. Catenin Alpha 2 Protein (Encoded by CTNNA2) as a Potential Drug Target in Multiple Psychiatric Disorders or Multimorbid Conditions

Although having a prominent role in intrauterine stages, *CTNNA2* is also expressed in the adult brain, showing different levels in patient groups versus controls, underpinning its potential as a drug target for psychiatric disorders.

*CTNNA2* has shown a reduced postmortem brain expression in schizophrenic patients versus controls, investigated within the parvocellular region of the mediodorsal thalamic nucleus [34], which has connections with several cortical and sub-cortical structures [35]. Moreover, schizophrenic non-smokers had a reduced hippocampal *CTNNA2* expression post mortem compared with schizophrenic smokers or mentally healthy non-smokers [36].

Catenin alpha 2 protein (encoded by *CTNNA2* gene in humans and *Ctnna2* in mice) is a linker between cadherin adhesion receptors and the actin cytoskeleton, and mice lacking its gene showed abnormally motile dendritic spine heads in the synapses of hippocampal neurons [37]. Its overexpression in these cells, however, restored the normal morphology of dendritic spines, and stabilized it over time [37], suggesting a possible therapeutic use in genetically vulnerable subgroups.

Pointing to the behavioral correlates and the possibility of a successful intervention in the case of a genetic vulnerability, the deletion of a gene partly overlapping *Ctnna2* in mice resulted in a larger startle to noise, an impaired pavlovian conditioned fear, and a weaker pre-pulse inhibition, all of which could be rescued by expressing *Ctnna2* in such mice [38].

Although the human adult expression of *CTNNA2* has been investigated only in schizophrenia, and pre-pulse inhibition is an endophenotype of schizophrenia [39], our results extend the relevance of *CTNNA2* in a diverse set of psychiatric symptoms and also in a transdiagnostic endophenotype—rumination.

Thus, in general, increasing the expression of *CTNNA2* or increasing the function of its catenin alpha 2 protein may have therapeutic potential in several psychiatric symptoms and perseverative negative thinking.

### 3.4. Limitations

Our study has several significant limitations. First, our cross-sectional design and the uncovered statistical mediation could only suggest the endophenotypic nature of rumination between *CTNNA2* and several psychiatric symptoms, but a longitudinal design would be needed to prove its causal role. Former studies on the possible endophenotypic role of rumination have also been cross-sectional; therefore, a longitudinal study would be crucial.

Second, our sample size entailed a very limited power to detect the effects of rare alleles or of weak effect sizes (Table 1). In our power calculations, the minimum minor allele frequency (MAF) was 0.01, and the hypothesized minimum effect size was β = 0.08. In case of all outcome variables, a larger sample would be needed to detect these kinds of effects. However, rs17019243 had a MAF of only 0.0489 in the sample of our mediation analyses, and had a standardized regression coefficient of 0.308 on RRS rumination in the model, suggesting enough power to detect effects in the case of not minimal but low MAFs and effect sizes.

## 4. Materials and Methods

### 4.1. Participants

This study is part of the Budakalasz Health Examination Survey (BHES), a cross-sectional voluntary cardiovascular screening program [40]. BHES was performed in 2011–2013 targeting the adult population (>20 years, around 8000 inhabitants) of a Central-Hungarian town (Budakalasz). The participation rate in the Budakalasz Health Examination Survey was approximately 30% (*n* = 2420) of the total eligible population.

Our present study included BHES participants with quality-controlled genomic data and non-missing phenotypic data on sex, age, rumination, and the investigated cardiovascular and psychiatric risk phenotypes (see below, in Section 4.4).

The study was approved by the Medical Research Council Scientific and Ethics Committee (approval no. 8224-0/2011/EKU [265/PI/11]). All of the procedures were done in accordance with the ethical standards of the responsible committee on human experimentation (institutional and national) and the Helsinki Declaration of 1975, as revised in 2000 (5). Written informed consent was obtained from all participants.

### 4.2. Measures

Medical history with special attention to cardiovascular disease (CVD)-related signs and symptoms, as well as lifestyle, was recorded by an experienced physician. Medical history was regarded as positive for hypertension if documented. For previously unknown hypertension, the following cut-off points were used: >140 mmHg systolic and/or >90 mmHg diastolic blood pressure. Current and former regular smokers were both considered smokers.

Anthropometric parameters (height and weight), rounded to the nearest 0.1 cm and 0.1 kg, were measured in a standing position while participants were wearing light indoor clothing without shoes. BMI was calculated with the Quatelet’s form [41]. Routine laboratory tests were performed in our Institution’s Central Laboratory with rigorous quality control. The concentration of lipid fractions was measured by using a colorimetric assay (Roche Diagnostics Ltd., Mannheim, Germany).

The ten years risk for developing CVD, coronary heart disease (CHD), myocardial infarction or coronary death (hard coronary heart disease (HCHD)), and stroke incidences were calculated for each participant using the relevant Framingham equations [42,43,44]. In detail, the Framingham-CVD score indicates a risk for any cardiovascular disease, and includes age, diabetes, smoking status, treated and untreated systolic blood pressure, total cholesterol, and high-density lipoprotein cholesterol (HDL-C) categories. Framingham-CHD indicates a risk score for coronary heart disease, and consists of gender, age, diabetes, smoking, blood pressure, total cholesterol categories, and low-density lipoprotein cholesterol (LDL-C) categories. Framingham-HCHD comprises age, total cholesterol, HDL-C, systolic blood pressure, treatment for hypertension, and smoking status. Framingham-stroke risk score includes age, systolic blood pressure, diabetes mellitus, cigarette smoking, prior CVD, atrial fibrillation, left ventricular hypertrophy, and use of hypertensive medication. The range of Framingham scores can be between 0 and 100%. In general, individuals with a low risk have 10% or less risk at 10 years, with intermediate risk of 10–20%, and with high risk of 20% or more.

To assess a wide range of psychiatric symptoms, the 53-item Brief Symptom Inventory (BSI) was used [45]. In addition to the global severity index (GSI) comprising all of the items, BSI has nine specific scales. These are somatization, obsessive-compulsive, interpersonal sensitivity, depression, anxiety, hostility, phobic anxiety, paranoid ideation, and psychoticism. Answers could be rated on a five-point Likert scale (0–4) for each item. Additional items that had not loaded straightforwardly on any factor in the original exploratory study were added to the depression scale in our study, because of their content.

Rumination was measured by the 10-item Ruminative Response Scale (RRS) [25], rated on a four-point Likert scale. Five items assessed the maladaptive brooding subtype, and the other five items belonged to the more adaptive reflection subtype.

In the case of each of the ten BSI scales and the three RRS scales, the sum score was divided by the number of items the participant responded to on that scale, and this mean score was used in all of the analyses.

Age was registered as the difference between birth date and the date of either medical examination or questionnaire filling, in years, rounded to one decimal.

### 4.3. Genotyping and Quality Control

Genotyping was performed using the Axiom Precision Medicine Research Array chip (https://www.thermofisher.com/order/catalog/product/902981#/902981 accessed on: 16 February 2021). Raw genotyping results were filtered using a rigorous quality control process, which followed the protocol in [46] and consisted of the following steps. First, biallelic, single-nucleotide variants with a minor-allele frequency (MAF) not less than 0.01 were retained. Then, variants and samples with a missing rate greater than 0.01 were excluded. This step was performed in an iterative manner, gradually increasing the missingness boundary from 0.1 through 0.05 to 0.01, at each step filtering genotypes first, and then the samples. Then, the genotypes not in Hardy–Weinberg equilibrium were discarded (i.e., the ones for which Plink’s “--hwe” test yielded a *p*-value less than 10^−5^). The SNPs remaining after this last step were kept for analysis, however the samples were further filtered.

For this latter part, the first a set of independent SNPs was selected using Plink’s “--indep-pairwise” command, using a sliding window of 1500 variants with a step size of 150 variants and with a linkage disequilibrium (LD) boundary of R^2^ ≤ 0.2. The samples for which the calculated heterozygosity F-score differs from the population-level mean by more than 3.0 standard deviations were excluded.

For analyses that require unrelated individuals, an identical-by-descent (IBD) filtering, with a threshold of 0.1875 and keeping only one member of the closely related pairs, was applied before the final heterozygosity-based filtering step.

This QC process resulted 274 SNPs within a ±10 kilobase extended boundaries of *CTNNA2* gene, according to genome build GRCh37/hg19.

### 4.4. Analyses

Descriptive statistics and associations between the phenotype measures were calculated using IBM SPSS v25 for the sample of 795 individuals from the primary analyses (see below). This sample consisted of 317 males and 478 females. To control for the variance of confounding factors, standardized residuals were used for association testing between the phenotypes, as follows. RRS rumination was regressed on sex and age. RRS brooding was regressed on sex, age, and RRS reflection. Finally, RRS reflection was regressed on sex, age, and RRS brooding. All three linear regression models were calculated with an enter method. The standardized residuals of each model were then tested for Pearson correlation with the five cardiovascular risk scores and the ten BSI scores.

For the primary analyses, a linear mixed modeling (LMM) approach, FaST-LMM [47], was used for a sample of 795 participants who had genetic data and non-missing phenotypic data for sex, age, BMI, Framingham scores, and BSI and RRS scores. The LMM with each SNP was run in an additive model for each of BMI, the four Framingham scores, the ten BSI scores, and the three RRS scores as the outcome variable. All of the models covaried sex, and a kinship matrix was included to handle the relatedness among participants and population stratification. This kinship matrix was calculated using the LOCO (leave-one-chromosome-out) method, meaning that kinship was estimated based on the whole genomic data, except for chromosome 2 (where *CTNNA2* resides). Age was also a covariate in each model, registered at the time point of either the medical examination (in case of BMI and Framingham scores) or the questionnaire filling (in case of BSI and RRS scores). For the RRS subscales of brooding and reflection, the other subscale was included as an additional covariate. Accounting for the LD between SNPs, we considered 225 independent (effective) SNPs behind our 274 SNPs, according to Gao et al.’s method [48]. The 18 models with these 225 effective SNPs resulted in 4050 tests and, consequently, a 1.235 × 10^−5^ Bonferroni-corrected significance threshold.

Power analyses for these primary tests were performed with Quanto v1.2 [49]. The means and standard deviations used for the calculations are detailed in Table 1 for each outcome variable. For every outcome variable, the type I error rate was set to a two-sided 0.05, and the minor allele frequency (MAF) range was hypothesized between 0.01 and 0.5 by 0.01. Based on previous studies, the effect size was expected to be β = 0.08 [9] or β = 1.62 [7] in an additive model, for every variable. The range of potential MAF values and the two hypothesized effect sizes resulted in a range of statistical power values to detect the *CTNNA2* SNP effects.

Secondary analyses were aimed at testing the mediating role of rumination in the significant findings of the primary analyses. For this purpose, we selected SNPs fulfilling the criteria of showing nominally significant (*p* ≤ 0.05) associations with both an RRS score and a cardiovascular or psychiatric risk score that had been significantly related to that particular RRS score according to the descriptive statistics. First, LMM was run with each of these SNPs for the particular cardiovascular or psychiatric risk score as an outcome, with the same method as in primary analyses, but including the particular RRS score as an additional covariate.

Second, from these SNPs shared between an RRS score and a disease risk score, we selected those that had been significantly associated with most of the disease risk scores and at least one RRS score in the primary analyses. To further filter these SNPs according to the statistical methods other than separate regression equations for each outcome, we performed a Bayesian Multilevel Analysis (BMLA), which treated all relationships between all variables in one complex model. For the BMLA, continuous phenotypic variables were discretized by their terciles, and we used a sample of unrelated individuals (*n* = 629) who had non-missing data on all of the included variables. Within the BMLA, a Markov Chain Monte Carlo method was used to estimate the a posteriori probabilities of the edges between the single variables in the Bayesian network with 1,000,000 burn-in steps and 5,000,000 sampling steps [50,51]. Using the same method, the categorical posterior probabilities were calculated for the BSI, RRS, and Framingham categories, which incorporated the corresponding variables. We used the Cooper–Herskovits priors [52] and the maximum number of parent connections were set to 4. Sex, age, and the selected SNPs were treated as exogenous variables, meaning that they could fulfill only causal roles in the model. Note that an a posteriori probability signifies the chance that a given relationship is present between the variables; thus, for example, a value above 0.5 means that a connection between variables is more probable than not.

SNPs showing strong a posteriori probabilities for both the particular RRS variables and the particular disease variables were selected for mediation testing.

Mplus v7.4 (https://www.statmodel.com/ accessed on: 14 July 2020) was used to test the statistical mediation of the particular RRS score between the SNP and the risk scores in structural equation modeling (SEM), which also hypothesize direct and indirect effects in the same model. An MLR (maximum likelihood estimation with robust standard errors) method was used within a sample of unrelated individuals (*n* = 638) having genetic data and non-missing phenotypic data on sex, age, RRS scores, and all the risk scores investigated in these secondary analyses. As, in Mplus, we could not enter a genome-wide kinship matrix to correct for relatedness and population stratification, only unrelated individuals were included, and the top ten principal components (PCs) of the genome were calculated to correct for potential stratification of the population. A PC was included in the SEM model if correlated with an outcome variable.

## 5. Conclusions

Our study suggests that the identified SNP of the *CTNNA2* gene is associated with rumination, and this type of perseverative negative thinking mediates a wide variety of psychopathologies and psychiatric disorders. These data strongly underscore the neurobiological basis of psychiatric disorders and point to the necessity of a new classification system of psychiatric disorders that was strongly suggested, but finally dropped at the construction of DSM-5.

Our data suggest the biomarker role of this gene, particularly its rs17019243 SNP, acting most likely through the *CTNNA2* expression levels, for rumination independently of the psychiatric diagnosis. Consequently, our results may help to develop new perspectives within personalized medicine and the stratification of highly heterogeneous patient groups into more biologically homogeneous subgroups.

Furthermore, increasing the expression or function of the catenin alpha 2 protein may be a new therapeutic approach in psychiatric patients, especially with rumination in their background.

Another new, more general approach could be the development of a possible new antiruminative agent that could serve as a psychiatric diagnosis independent (trans-psychopharmacon) medicine in psychiatry. As rumination may precede depression and PTSD, the treatment of rumination may also serve as a preventive medication in depression, schizophrenia, and other psychiatric disorders in this specific ruminating population.

## Figures and Tables

**Figure 1 pharmaceuticals-14-00850-f001:**
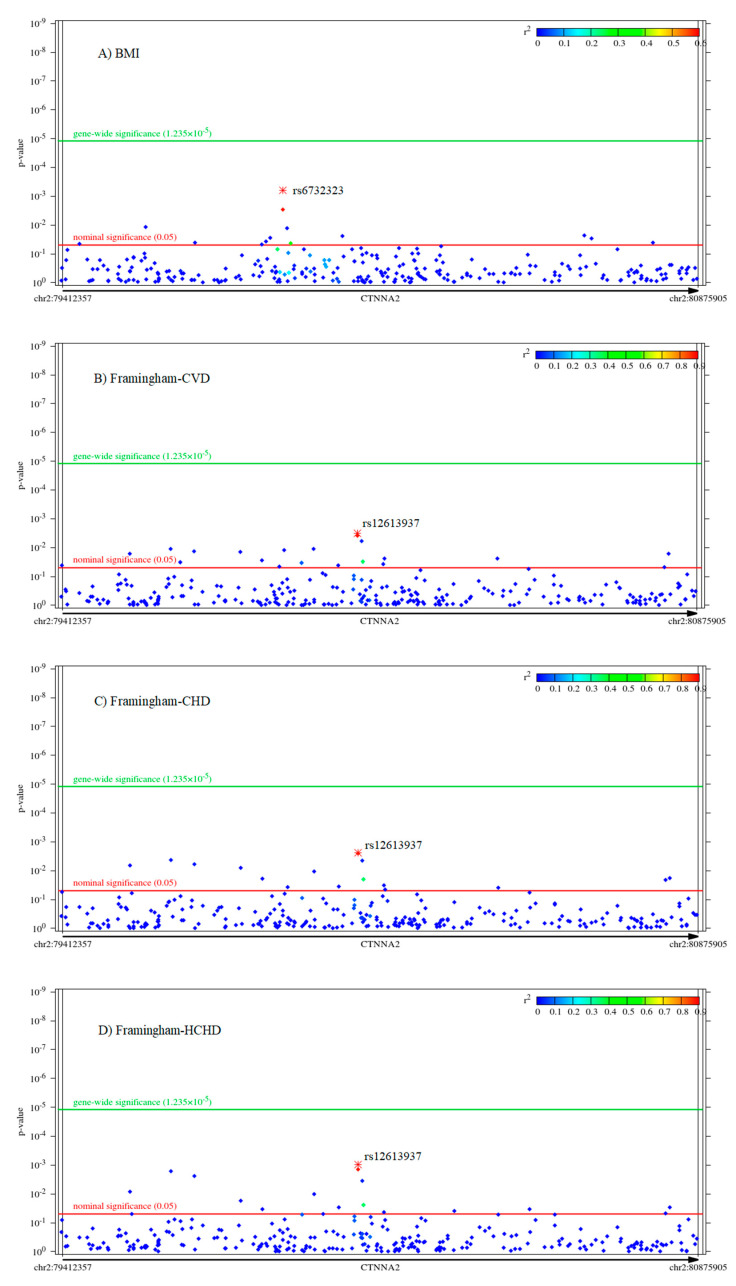
Manhattan plots for the associations of each *CTNNA2* SNP with each outcome variable. (**A**–**E**) display cardiovascular risk phenotypes, (**F**–**O**) display BSI psychiatric symptom scores, and (**P**–**R**) display RRS rumination, brooding and reflection scores, respectively. The *p*-value is displayed with two thresholds, Bonferroni-corrected gene-wide and nominal significance, as a function of the SNP position within the gene. We can see that, while no SNP survived the gene-wide significant threshold, there are nominally significant associations for all 18 phenotypes. R^2^—linkage disequilibrium with the most significant SNP (marked with asterisk and given by name); SNP—single-nucleotide polymorphism; chr—chromosome; BMI—body mass index; CVD—cardiovascular disease; CHD—coronary heart disease; HCHD—hard coronary heart disease: myocardial infarction or coronary death; BSI—Brief Symptom Inventory; RRS—Ruminative Response Scale.

**Figure 2 pharmaceuticals-14-00850-f002:**
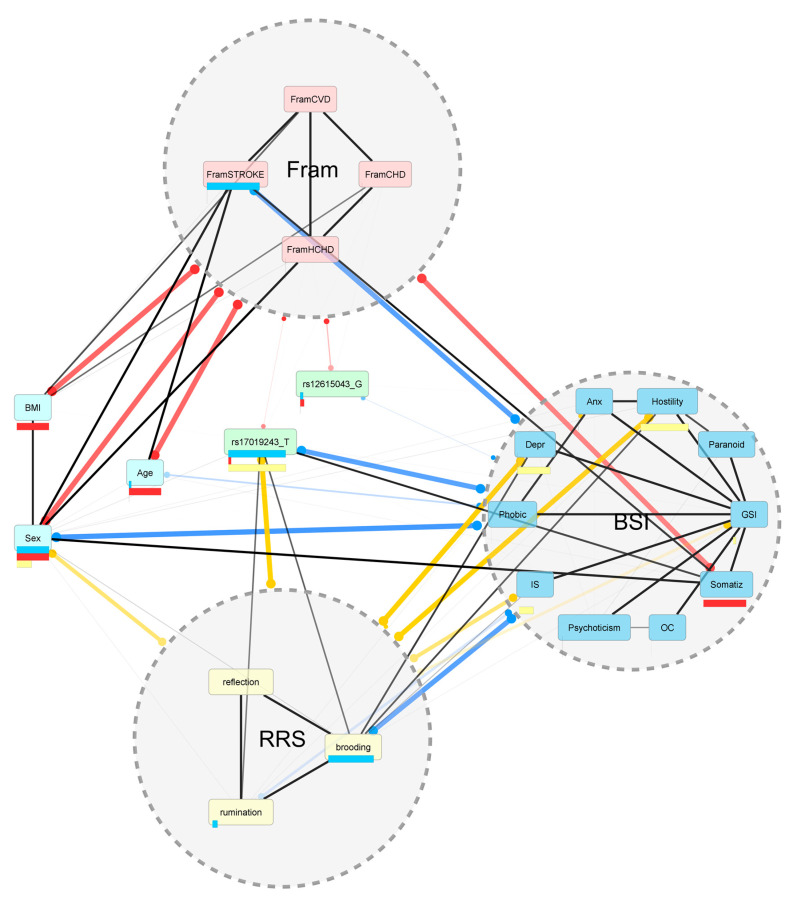
The resulting network model from the Bayesian Multilevel Analysis. The edges (all lines) represent a probabilistic relationship between the variables, namely single variables and categories of variables. Single variables include the 18 phenotypes, sex, age, and the two SNPs. Categories of variables include Framingham, RRS, and BSI. The thickness of the edges is proportional with the posterior probability, signifying a more probable real-life connection between the variables (numerical posterior probability values are detailed in Appendix A). The black edges connect single variables with each other, and the colored edges connect single variables with the categories, where the color encoding is as follows: red—Framingham; yellow—RRS; blue—BSI. The length of the horizontal bars below the nodes represents the posterior probability values with respect to the categories. The results suggest a real-life relationship between rs17019243 and RRS, and BSI single variables and categories as well. However, rs12615043 has no real-life relationship with any of them. RRS—Ruminative Response Scale; BSI—Brief Symptom Inventory; BMI—body mass index; CVD—cardiovascular disease; CHD—coronary heart disease; HCHD—hard coronary heart disease: myocardial infarction or coronary death; GSI—global severity index; anx—anxiety; depr—depression; IS—interpersonal sensitivity; OC—obsessive-compulsive.

**Figure 3 pharmaceuticals-14-00850-f003:**
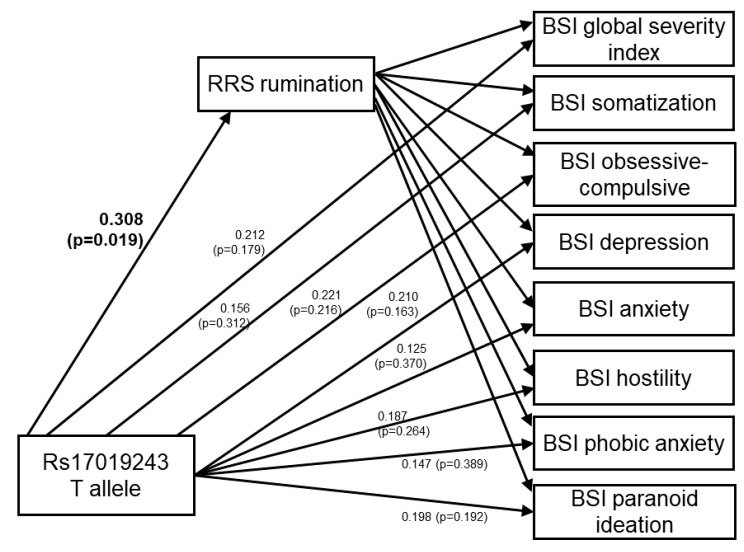
Structural equation modeling to test the direct and indirect effects of *CTNNA2* rs17019243 for BSI psychiatric symptom scores via RRS rumination within the same model. Model fit indices: χ^2^ = 20.184 (df = 14; *p* = 0.1244); RMSEA = 0.026 (PCLOSE = 0.949); CFI = 0.999; TLI = 0.992. Direct effects of rs17019243 are displayed in the figure, and the significant (*p* ≤ 0.05) one is marked in bold. The results suggest that rs17019243 has a direct effect on RRS rumination, but has no direct effect on any of the BSI scores. BSI—Brief Symptom Inventory; RRS—Ruminative Response Scale; df—degrees of freedom; RMSEA—root mean squared error of approximation; PCLOSE—significance of the statistical hypothesis that RMSEA is different from the desirable ≤0.05; CFI—comparative fit index (>0.90 indicates a good model fit, and >0.95 indicates a very good model fit); TLI—Tucker−Lewis index (>0.90 indicates a good, and >0.95 a very good model fit).

**Table 1 pharmaceuticals-14-00850-t001:** Descriptive statistics for the measures of our study (*n* = 795). Range of Framingham scores can be between 0 and 100%, BSI scores can range from 0 to 4, and RRS scores from 1 to 4. The results suggest that our study sample is generally overweighted and has an intermediate risk for CVD, but a low risk for specific CVD diseases. We can also see a wide range of statistical power values to detect *CTNNA2* SNP effects on all phenotypes, and good internal consistencies for psychometric scales. BMI—body mass index; CVD—cardiovascular disease; CHD—coronary heart disease; HCHD—hard coronary heart disease: myocardial infarction or coronary death; BSI—Brief Symptom Inventory; RRS—Ruminative Response Scale.

Variable	Mean	Standard Error of Mean	Standard Deviation	Power to Detect *CTNNA2* SNP Effect	Cronbach’s Alpha
Age at medical examination	53.72	0.490	13.815		
Age at questionnaire filling	54.85	0.500	14.109		
BMI	27.39	0.179	5.034	5.05–99.99%	
Framingham-CVD	13.78	0.426	11.998	5.01–76.98%	
Framingham-CHD	8.59	0.277	7.821	5.02–98.58%	
Framingham-HCHD	4.17	0.181	5.097	5.04–99.99%	
Framingham-stroke	2.83	0.117	3.288	5.11–99.99%	
BSI global severity index	0.55	0.018	0.508	9.58–99.99%	0.957
BSI somatization	0.51	0.022	0.617	8.08–99.99%	0.800
BSI obsessive-compulsive	0.70	0.025	0.703	7.37–99.99%	0.828
BSI interpersonal sensitivity	0.94	0.025	0.695	7.42–99.99%	0.643
BSI depression	0.48	0.021	0.606	8.20–99.99%	0.862
BSI anxiety	0.54	0.024	0.668	7.62–99.99%	0.824
BSI hostility	0.50	0.020	0.566	8.67–99.99%	0.733
BSI phobic anxiety	0.32	0.021	0.584	8.45–99.99%	0.800
BSI paranoid ideation	0.71	0.024	0.673	7.59–99.99%	0.737
BSI psychoticism	0.42	0.020	0.564	8.70–99.99%	0.675
RRS rumination	1.89	0.018	0.503	9.67–99.99%	0.812
RRS brooding	1.91	0.020	0.570	8.62–99.99%	0.759
RRS reflection	1.86	0.021	0.602	8.24–99.99%	0.753

**Table 2 pharmaceuticals-14-00850-t002:** Pearson correlations of RRS score residuals with cardiovascular and psychiatric risk scores (*n* = 795). Significant (*p* ≤ 0.05) correlations are marked in bold. Among the cardiovascular risk scores, only BMI is correlated, and only weakly, with RRS scores: positively with brooding and negatively with reflection. RRS rumination and brooding are correlated moderately with all ten BSI scores, but reflection has a weak correlation with obsessive-compulsive and depressive symptoms. *p*—*p*-value; BMI—body mass index; CVD—cardiovascular disease; CHD—coronary heart disease; HCHD—hard coronary heart disease: myocardial infarction or coronary death; BSI—Brief Symptom Inventory; RRS—Ruminative Response Scale.

Variable		RRS Rumination Score Residual	RRS Brooding Score Residual	RRS Reflection Score Residual
RRS rumination score residual	Pearson correlation	1	**0.508**	**0.532**
	*p*		**2.148 × 10^−53^**	**2.291 × 10^−59^**
RRS brooding score residual	Pearson correlation	**0.508**	1	**−0.459**
	*p*	**2.148 × 10^−53^**		**1.263 × 10^−42^**
RRS reflection score residual	Pearson correlation	**0.532**	**−0.459**	1
	*p*	**2.291 × 10^−59^**	**1.263 × 10^−42^**	
BMI	Pearson correlation	−0.017	**0.079**	**−0.095**
	*p*	0.632	**0.026**	**0.007**
Framingham-CVD	Pearson correlation	0.019	0.020	−0.001
	*p*	0.598	0.565	0.982
Framingham-CHD	Pearson correlation	0.015	0.011	0.005
	*p*	0.667	0.756	0.890
Framingham-HCHD	Pearson correlation	0.027	0.017	0.011
	*p*	0.451	0.636	0.754
Framingham-stroke	Pearson correlation	0.024	0.020	0.005
	*p*	0.502	0.568	0.896
BSI global severity index	Pearson correlation	**0.454**	**0.435**	0.040
	*p*	**1.357 × 10^−41^**	**4.553 × 10^−38^**	0.258
BSI somatization	Pearson correlation	**0.278**	**0.299**	−0.007
	*p*	**1.586 × 10^−15^**	**7.339 × 10^−18^**	0.835
BSI obsessive-compulsive	Pearson correlation	**0.364**	**0.301**	**0.080**
	*p*	**2.272 × 10^−26^**	**4.043 × 10^−18^**	**0.024**
BSI interpersonal sensitivity	Pearson correlation	**0.355**	**0.398**	−0.026
	*p*	**5.627 × 10^−25^**	**1.259 × 10^−31^**	0.471
BSI depression	Pearson correlation	**0.456**	**0.403**	**0.074**
	*p*	**4.600 × 10^−42^**	**1.834 × 10^−32^**	**0.037**
BSI anxiety	Pearson correlation	**0.404**	**0.384**	0.039
	*p*	**1.355 × 10^−32^**	**2.087 × 10^−29^**	0.271
BSI hostility	Pearson correlation	**0.334**	**0.343**	0.007
	*p*	**4.033 × 10^−22^**	**2.014 × 10^−23^**	0.852
BSI phobic anxiety	Pearson correlation	**0.302**	**0.276**	0.040
	*p*	**2.952 × 10^−18^**	**2.199 × 10^−15^**	0.256
BSI paranoid ideation	Pearson correlation	**0.363**	**0.395**	−0.013
	*p*	**3.710 × 10^−26^**	**5.287 × 10^−31^**	0.707
BSI psychoticism	Pearson correlation	**0.368**	**0.346**	0.040
	*p*	**6.039 × 10^−27^**	**8.015 × 10^−24^**	0.266

**Table 3 pharmaceuticals-14-00850-t003:** Standardized estimates of the indirect effects of *CTNNA2* rs17019243 on each BSI score via RRS rumination. All mediated effects are significant at a *p* ≤ 0.05. We can see that rs17019243 acts on eight BSI scores via RRS rumination, with the strongest effect on GSI, depression, anxiety, obsessive-compulsive, and paranoid ideation scores. RRS—Ruminative Response Scale; BSI—Brief Symptom Inventory; GSI—global severity index.

*CTNNA2* rs17019243 → RRS Rumination → BSI Score							
GSI	Somatization	Obsessive- Compulsive	Depression	Anxiety	Hostility	Phobic Anxiety	Paranoid Ideation
0.146	0.098	0.117	0.146	0.133	0.107	0.098	0.114
(*p* = 0.020)	(*p* = 0.023)	(*p* = 0.020)	(*p* = 0.021)	(*p* = 0.020)	(*p* = 0.022)	(*p* = 0.022)	(*p* = 0.023)

## Data Availability

The data presented in this study are available upon request from the corresponding author. The data are not publicly available due to ethical considerations.

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
