# Peer review of "Catenin Alpha 2 May Be a Biomarker or Potential Drug Target in Psychiatric Disorders with Perseverative Negative Thinking"

_pharmaceuticals, 2021, doi:10.3390/ph14090850_

Round 1
Reviewer 1 Report
The manuscript by Eszlari et.al. is an important study considering it established CTNNA2 genetics and/or expression levels as biomarkers and therapeutic approaches in psychiatric disorders with perseverative negative thinking including e.g., including depression. The manuscript is well-written, with a comprehensive presentation of results and discussion, which makes the impact of the study high. Please see below for some minor comments that can substantiate the paper even more.
- The authors need to clearly rationalize the hypothesis they intend to test in the study. The two arms of the hypothesis are very disconnected which makes it hard to understand the foundation upon which the study is designed.
- It is not clear why did the authors test the role of SNP occurring in the gene CTNNA2?
- The authors should clearly state why the 18 stated phenotypes were selected for assessment of psychiatric risk scores? Do they believe that the selection of other phenotypes can affect the outcome of the association study presented here?
- SNPs occurrence is independent of the tissue-based expression. On what ground do the authors discuss that "Another possible explanation is that CTNNA2 has pleiotropic effects on rumination and cardiovascular phenotypes even at the level of tissues and cell types."?
- It is not very clear from the results or even the discussion, how do the authors propose CTNNA2 as a potential biomarker for psychiatric disease, based on post-mortem analysis of the protein from the targeted brain region? Is there any report of changed expression of CTNNA2 from CSF, blood, plasma, or other biofluids in these described psychiatric states?
- Rumination versus Brooding - this is a critical piece of assumption based on which certain interpretations have been made in the data presented. The authors should have a detailed description of what the difference is between rumination and brooding, and also why is it important for this to be considered for the study?
Reviewer 2 Report
Original, sound research, I would probably try and shorten the length of the article. Best of luck for your future research aswell.
